# The Effectiveness and Safety of Acupuncture on Suicidal Behavior: A Systematic Review

**DOI:** 10.3390/healthcare11070955

**Published:** 2023-03-27

**Authors:** Chan-Young Kwon, Boram Lee

**Affiliations:** 1Department of Oriental Neuropsychiatry, College of Korean Medicine, Dong-eui University, 52-57, Yangjeong-ro, Busanjin-gu, Busan 47227, Republic of Korea; 2KM Science Research Division, Korea Institute of Oriental Medicine, 1672, Yuseong-daero, Yuseong-gu, Daejeon 34054, Republic of Korea

**Keywords:** acupuncture, suicide, suicide prevention, suicidal behavior, systematic review

## Abstract

In situations where death by suicide is a major global issue and effective prevention and management approaches are lacking, acupuncture improves some risk factors for suicide, including depression, and it has been used for a long time in clinical settings. Herein, we aimed to assess the effectiveness and safety of acupuncture in the treatment of suicidal behaviors. Fourteen electronic databases were searched for studies published up to 7 September 2022. Original interventional studies of acupuncture in suicide prevention were included. The primary outcome was the validated measure of suicidal ideation. The risk of bias in the included studies was assessed using an appropriate assessment tool. Due to the heterogeneity of the included studies, only qualitative analyses were conducted. Eight studies on manual acupuncture (50%), electro-acupuncture (37.5%), and acupressure (12.5%) were included. In particular, three studies (37.5%) used the National Acupuncture Detoxification Association protocol to stimulate the bilateral sympathetic, Shenmen, kidney, liver, and lung auricular points. Acupuncture was effective in direct and indirect outcomes related to suicidal behavior, not only for participants with suicidal behavior, but also for those with other conditions, including depression. A decrease in salivary cortisol was the only biological indicator of acupuncture in patients with suicidal ideation. However, the methodological quality of the included studies was not optimal. In conclusion, acupuncture may reduce the risk of suicidal behavior in clinical and non-clinical populations. Owing to clinical heterogeneity, low methodological quality, and the small number of included studies, further high-quality studies should assess the effectiveness of acupuncture.

## 1. Introduction

Suicide is a major global health problem, with approximately 790,000 suicide deaths reported worldwide in 2019, according to a study that analyzed the mortality database of the World Health Organization (WHO) and the Global Burden of Disease Study [1]. Additionally, according to an analysis of the WHOs World Mental Health Surveys, the estimated 12-month prevalence of suicidal thoughts, plans, and attempts in developed countries was 2.0%, 0.6%, and 0.3%, respectively [2]. Worldwide, suicide accounts for 1.4% of all premature deaths, the majority of which are related to mental illnesses, particularly depressive disorders, substance use, psychosis, post-traumatic stress disorder (PTSD), and impulse control [3,4]. However, risk factors for suicide are not limited to mental illness. For example, some physical health conditions such as human immunodeficiency virus infection and acquired immunodeficiency syndrome, sleep disorders, and traumatic brain injury are also associated with a high risk of suicide [5]. Additionally, social context, including unjust and discriminatory social conditions and attitudes, may contribute to an individual’s suicide [6]. Common methods of suicide attempt include hanging, self-poisoning with pesticides, and the use of firearms [3]. Therefore, suicide prevention should consider complex factors at both individual and social levels.

Some psychological interventions (cognitive behavioral therapy and dialectical behavior therapy) and pharmacological treatments (ketamine, lithium, and clozapine) are associated with a reduced risk of suicide; however, evidence is limited [7]. Moreover, the benefit of antidepressants in preventing suicidal behavior is highly controversial in some vulnerable populations, such as children and adolescents [8]. The most comprehensive review conducted to date has highlighted the importance of combinations of evidence-based strategies over single preventive interventions in suicide prevention [9]. Suicide prevention strategies should be applied not only at the individual level but also at the population level [9], and suicide prevention strategies at the national level must be introduced [10]. Therefore, to address the national suicide problem, a comprehensive review of the benefits and risks of the available multidisciplinary evidence-based strategies is required.

The primary care approach is as important as a multidisciplinary approach to suicide prevention. It is well known that suicide victims are overwhelmingly more likely to visit primary care facilities than mental health services before the completion of suicide [11]. In this context, primary care is an ideal setting for identifying individuals at risk of suicide and initiating mental health treatment [12]. East Asian traditional medicines (EATMs), such as traditional Chinese medicine, serve as primary care in some countries [13]. In others, the proportion of individuals using complementary and integrative medicine (CIM) for mental health problems is high [14]. Therefore, investigating the applicability of the EATM or CIM modality for suicide prevention from multidisciplinary evidence-based strategies, as well as the primary care perspective, is considered a reasonable strategy. Additionally, given the differences in mortality statistics across countries and the importance of developing programs suited to each country’s policy [3], it may be more relevant in countries where the use of either the EATM or CIM modality is common.

Acupuncture, a representative EATM or CIM modality, is a procedure that includes needle stimulation of the acupoint as well as close practitioner–patient communication [15]. Acupuncture has been shown to have therapeutic benefits for the known risk factors for suicidal behaviors, including depression [16], PTSD [17], and chronic pain [18]. However, the role of acupuncture as a preventive or therapeutic strategy for suicidal behavior has not yet been critically evaluated from the perspective of evidence-based medicine. Therefore, this review aimed to investigate the effectiveness and safety of acupuncture for suicidal behaviors. 

## 2. Materials and Methods

The protocol of this systematic review has been registered in international registries (Open Science Framework: 3EKN9; PROSPERO: CRD42022334375) and published in a peer-reviewed journal [19]. No amendments were issued after the protocol’s registration. This systematic review was conducted in accordance with the Preferred Reporting Items for Systematic Reviews and Meta-Analysis statements [20] (Appendix A).

### 2.1. Data Sources and Search Strategy

The electronic databases in which a comprehensive search was conducted for this systematic review included MEDLINE (via PubMed), the Cochrane Central Register of Controlled Trials, EMBASE (via Elsevier), Allied and Complementary Medicine Database (via EBSCO), PsycARTICLES (via ProQuest), Cumulative Index to Nursing and Allied Health Literature (via EBSCO), China National Knowledge Infrastructure, Wanfang Data, VIP Chinese Science and Technology Periodicals, Citation Information by NII, Korean Studies Information Service System, Korea Citation Index, Research Information Sharing Service, Oriental Medicine Advanced Searching Integrated System, and Korean Medical database. All relevant studies published by the search date of 7 September 2022 were included. In addition to database searches, reference lists of relevant review articles were reviewed to identify potentially missing literature. Additionally, manual searches were conducted using Google Scholar. The search strategy and results for each database are presented in Appendix A.

### 2.2. Eligibility Criteria

The inclusion criterion for this systematic review was the P-I-C-O-S format, as previously described [19]. (1) Population: Although the risk of suicidal behavior is relatively prevalent in the clinical population, including individuals with depressive disorders, suicidal behavior is still found in the non-clinical population; therefore, this systematic review did not limit the population standard. Both clinical and nonclinical populations were included in this study. However, participants with suicidal behavior and others were analyzed separately. No restrictions were placed on the sex, age, ethnicity, or race of the participants. (2) Intervention: All acupuncture treatments were included in this review. Acupuncture refers to a non-pharmacological treatment in which a tool such as a needle is inserted into a specific point (acupoint) of the body for stimulation by clinicians or practitioners. There were no restrictions on whether this intervention was used as a monotherapy or adjuvant therapy. Additionally, there were no restrictions on whether the needle penetrated the skin or not. However, acupoint herbal injections or pharmacopuncture injections of pharmacological agents into the acupoint were excluded. (3) Comparator: No restrictions were placed on the comparator conditions. That is, no treatment, waitlist, sham control, or active comparators, such as antidepressants, were allowed. (4) Outcome: The primary outcome of this review was validated measures of suicidal ideation, including the Beck Scale for Suicidal Ideation (BSSI) [21]. Any other measures of suicidal behavior, including suicidal ideation, attempts, or completion, were considered secondary outcomes. Among the included studies, in the case of studies targeting participants with suicidal behavior, all the reported outcomes were analyzed, not limited to suicidal behavior. (5) Study type: Any original interventional study was allowed. Randomized controlled clinical trials (RCTs), non-randomized controlled clinical trials (CCTs), and before-and-after studies were all allowed. However, retrospective studies, observational studies, reviews, and preclinical studies were excluded. No restrictions were placed on the language of publication or publication status. Thus, gray literature, such as conference abstracts and dissertations, were also allowed.

### 2.3. Study Selection Process

The study selection was conducted in two stages. The initially searched documents were reviewed for their titles and abstracts, and potentially relevant studies were selected through this first step. Documents that were difficult to assess for their potential relevance were passed to the second step. In the second step, the full texts of the documents were reviewed to determine whether they met the inclusion criteria. This two-step selection process was conducted by two independent authors (CY Kwon and B Lee), and any disagreements were resolved through a discussion. Third-party intervention was not required during the study selection process. EndNote20 (Clarivate Analytics, Philadelphia, PA, USA) was used to manage the bibliographic data.

### 2.4. Data Extraction Process

Data extraction was performed using a standardized, predefined, pilot-tested extraction method. This process was performed using Microsoft Excel 365 software (Microsoft, Redmond, WA, USA). The data extracted included the sample size, characteristics of the participants, types of experimental and control interventions, outcome measures and results, adverse events or safety profile, and information for the assessment of the methodological quality of the included study. The extraction process was conducted by two independent authors (CY Kwon and B Lee), and the extracted forms were collated. Disagreements were resolved through a discussion, and no third-party intervention was necessary. In case of missing or uncertain data, the first and/or corresponding author of the document was contacted via e-mail.

### 2.5. Quality Assessment Process

The methodological quality of RCTs among the included studies was assessed using the risk of bias tool-2 [22] developed by the Cochrane Group. This tool evaluates the potential bias of the RCTs in the following five domains and derives an overall bias based on the evaluation: bias arising from the randomization process, bias due to deviations from intended interventions, bias due to missing outcome data, bias in the measurement of the outcome, and bias in the selection of the reported result [22]. Each domain item is classified as “high risk”, “some concerns”, or “low risk” [22]. The methodological quality of the CCTs was assessed using the Risk of Bias Assessment tool for Non-randomized Studies [23]. This tool evaluates the potential bias of non-randomized studies including CCTs in the following eight domains: the possibility of target group comparisons, target group selection, confounder, exposure measurement, blinding of assessors, outcome assessment, incomplete outcome data, and selective outcome reporting [23]. Each domain item is classified as “high risk of bias”, “unclear risk of bias”, and ”low risk of bias” [23]. In the case of before–after studies, the Quality Assessment Tool for Before–After (Pre–Post) Studies with No Control Group, evaluating 12 defined criteria, was used to assess the methodological quality of the studies [24]. The quality assessment process was conducted by two independent authors (CY Kwon and B Lee), and any discrepancies were resolved through a discussion.

### 2.6. Data Synthesis and Analysis Process

In the protocol of this review, quantitative synthesis, including a cumulative meta-analysis, was planned [19]. However, owing to the heterogeneity of the included studies, no quantitative synthesis was conducted in this review. Specifically, among the RCTs included in this review, no two or more studies reported the same outcome. All the included studies were analyzed qualitatively. This qualitative analysis was divided into categories of the population: participants with suicidal behavior and others. To analyze acupuncture in more detail, the intervention of interest and acupuncture procedures were analyzed in detail with reference to the revised Standards for Reporting Interventions in Clinical Trials of Acupuncture [25].

## 3. Results

### 3.1. Study Selection

In the initial search, 2516 articles were retrieved, including an additional 2 records identified through other sources. In the first step, the titles and abstracts of 2350 articles were reviewed, excluding duplicate articles. During the initial screening process, 2332 articles evaluated as not relevant to this systematic review were excluded. In the second step, through a full-text review, nine articles that did not meet the inclusion criteria were excluded. The reasons for the papers excluded in the second step are as follows: a case study of self-injury after acupuncture [26], a letter to the editor without clinical data [27], four observational studies [28,29,30,31], and three not reporting suicidal behaviors as their outcomes [32,33,34]. Finally, eight studies in nine articles [35,36,37,38,39,40,41,42,43] were included in this review by the two-step selection process. As Yang [36] (dissertation) and Yang et al. [42] (journal article) reported the same study, only the former [36] will be cited; however, some of its data derives from the latter [42]. The study selection process is illustrated in Figure 1.

### 3.2. Characteristics of Included Studies

All the included studies [35,36,37,38,39,40,41,42,43] were published between 2014 and 2021. A total of 351 participants were included in these studies. When classified by the study design, there were four RCTs [35,37,40,43], one CCT [36], one before–after study [38], and two single-case experimental studies [39,41]. As for the countries where the studies were conducted, China was the most common with five [35,36,37,38,43], followed by Iran [39,40,41]. Four studies [38,39,40,41] targeted participants with suicidal ideation, and the remaining four studies [35,36,37,43] targeted those with depression, although suicidal behavior was not specified in their inclusion criteria. The longest duration of treatment was 3 months [38]; however, 4 weeks or 1 month accounted for more than half [37,39,40,41,43] of the included studies. The characteristics of the included studies are summarized in Table 1.

### 3.3. The Methodological Quality of Included Studies

The methodological quality of two single-case experimental studies [39,40] could not be evaluated because of the absence of an appropriate tool to evaluate the quality of the study. All four RCTs [35,37,40,43] were assessed due to some concerns regarding bias owing to deviations from the intended interventions because they did not report information on whether deviations from the intended intervention occurred due to the clinical trial context. Additionally, because all studies [35,37,40,43] did not report whether data analysis was conducted according to a preliminary plan before the unblinding of the outcome data, all studies had some concerns about bias in the selection of the reported results. In particular, two studies [37,43] had a high risk of bias in the measurement of the outcome because outcome assessors were not blinded, and assessments could be influenced by knowledge of the intervention. Overall, two RCTs [37,43] were assessed to have a high risk of overall bias, and there were some concerns regarding another two RCTs [35,40] regarding overall bias. One CCT [36] was judged to have a high risk of selection bias because the absence of outcomes from the study participants was not confirmed at the time of enrollment, and the main confounders were not confirmed or considered properly during the planning and analysis stages. It had a low risk of performance bias, confirmation bias, and reporting bias; however, it had an unclear risk of attrition bias because it did not report the cause of missing data. One before–after study [38] clearly stated the study question and intervention and defined and assessed the outcome measures consistently across all the study participants. Additionally, the study [38] reported the recruitment of the study population. As the study [38] used self-reported questionnaires, the blinding of outcome assessors was not possible (Table 2).

### 3.4. Details of Acupuncture Procedure

Regarding acupuncture stimulation methods, manual acupuncture was used in four studies [38,39,40,41], electro-acupuncture in three [35,36,43], and acupressure in one study [37]. Four studies used auricular acupuncture points [37,39,40,41], and three of them [39,40,41] used bilateral sympathetic, Shenmen, kidney, liver, and lung acupuncture points of the National Acupuncture Detoxification Association (NADA) protocol. The remaining four studies [35,36,38,43] used body acupuncture points. Among the four studies [38,39,40,41] targeting participants with suicidal ideation, three studies used NADA protocol [39,40,41]. Only one study [41] reported the practitioner’s background, and acupuncture was performed by a trained physician and an acupuncturist with a certification and five-year history of the treatment. The needle or patch retention time varied from 30 min to 3 days, and acupuncture frequency was evenly distributed from once a week to seven times a week. For concurrent treatment, prescribed antidepressants [35]; conventional treatment for stroke, including blood pressure control, cerebral edema management, and rehabilitation training [37]; infrared irradiation [38]; methadone syrup [39,40,41]; and music therapy plus routine care [43] were used (Table 3).

### 3.5. Impact of Acupuncture on Subjects with Suicidal Ideation

#### 3.5.1. RCT (n = 1)

Pirnia et al. [40] recruited patients with dysthymia with suicidal thoughts who received methadone maintenance treatment. The participants were randomly divided into two groups. The treatment group (n = 12) received auricular acupuncture (the NADA protocol), and the control group (n = 12) received sham auricular acupuncture (1 cm away from the real acupoints) for a total of 8 sessions for 4 weeks. After treatment, the auricular acupuncture group had significantly lower salivary cortisol levels (*p* = 0.006) and BSSI scores (*p* = 0.01) than those in the control group (Appendix A).

#### 3.5.2. Before–After Study (n = 1)

Jin et al. [38] performed manual acupuncture for a total of 44 sessions over 3 months for patients (n = 21) with depression and suicidal ideation and found that the effective rate, defined as the reduction rate of HAMD, reached 90.48% (19/21). In addition, 66.67% of the participants (14/21) were free from suicidal ideation after treatment. The BSSI score showed a significant decrease (*p* < 0.05) (Appendix A).

#### 3.5.3. Single-Case Experimental Study (n = 2)

The research team who conducted an RCT [40] on patients with dysthymia with suicidal thoughts also reported two single-case experimental studies on patients with suicidal ideation [39,41]. In the studies [39,41], auricular acupuncture (NADA protocol) was performed in 2 sessions per week for 4 weeks, for a total of 8 sessions, on the patients. Routine care for the patients, including methadone syrup, was maintained during the auricular acupuncture. In one study of a patient with major depression and suicidal ideation [39], the reductions in salivary cortisol and suicidal ideation were observed after the treatment. In another study of a patient with prostate carcinoma and suicidal ideation, the scores of the Beck Depression Inventory (BDI) and BSSI, and salivary cortisol in the morning, were all significantly reduced (all, *p* < 0.05) (Appendix A).

### 3.6. Impact of Acupuncture on Participants with Other Conditions

#### 3.6.1. RCT (n = 3)

In the three-arm RCT by Wang [35], patients with a depressive episode were randomly divided into the electro-acupuncture and placebo pill group (group A, n = 20), electro-acupuncture and escitalopram group (group B, n = 21), and sham electro-acupuncture and escitalopram group (group C, n = 20). Electro-acupuncture or sham electro-acupuncture was administered for 6 weeks for a total of 24 sessions. The Colombia Suicide Rating Scale (CSRS) was used as a safety index to assess the suicide risk of the participants included in this study. Though the raw data of the CSRS were not reported, the authors reported that no cases of suicide risk occurred among participants during the study period when assessed using the CSRS. We contacted the corresponding author to request the missing raw data by e-mail; however, there was no response. In the three-arm RCT by Li et al. [37], patients with post-stroke depression were randomly divided into auricular acupuncture using a magnet piece group (group A, n = 31), auricular acupuncture by a vaccaria seed group (group B, n = 31), and a control group (group C, n = 31). All the participants were provided with a conventional treatment for stroke, including blood pressure control, cerebral edema management, and rehabilitation training. Two types of auricular acupuncture were performed for a total of 12 sessions over 4 weeks. At 4 weeks post-treatment, the suicide score evaluated by item 3 of the HAMD-17 was significantly lower in both A and C groups than in group C (both, *p* < 0.01). In addition, the score of group A was significantly lower than that of group B (*p* < 0.01). Li et al. [43] recruited patients with mild and moderate depression and randomly divided them into treatment (n = 45) and control (n = 45) groups. Electro-acupuncture and music therapy were administered to the treatment group, while only electro-acupuncture was administered to the control group. All participants received routine care, including psychological and health education. Electro-acupuncture was performed daily for 1 month. Although this study investigated the additive effect of music therapy, a statistically significant decrease in Nurses’ Global Assessment of Suicide Risk (NGASR) score was observed in the control group after treatment (*p* < 0.05). However, the group treated with music therapy had a significantly lower NGASR score after treatment than the control group (*p* < 0.05) (Appendix A).

#### 3.6.2. CCT (n = 1)

Yang [36] divided patients with depressive episodes into treatment and control groups. In the treatment group (n = 35), electro-acupuncture was administered for 8 weeks for a total of 24 sessions, and in the control group (n = 25), selective serotonin reuptake inhibitors were administered for the same period. Their outcomes included CSRS; however, the results were not reported in either document [36,42]. We contacted the corresponding author to request the missing raw data by e-mail; however, there was no response. In one study [42], it was described that none of the patients showed suicide risk during the trial period in the CSRS evaluation (Appendix A).

### 3.7. Safety of Acupuncture

Adverse event data related to acupuncture were reported in only three studies [35,36,37]. Wang [35] reported that among participants who received electro-acupuncture and placebo pill, two cases of nausea (2/19, 10.5%), two cases of fatigue (2/19, 10.5%), and one case of itchiness (1/19, 5.3%) were reported as adverse reactions. Yang [36] reported that among participants who received electro-acupuncture, bleeding and hematoma around the site of the needling occurred as adverse reactions in 12.5% (4/32) of cases. Li et al. [37] reported that there were no adverse reactions during the study period (Appendix A).

## 4. Discussion

### 4.1. Summary of Findings

This systematic review investigated the effectiveness and safety of acupuncture for suicidal behavior. Through a comprehensive search, eight studies from nine documents [35,36,37,38,39,40,41,42,43] were included in this review. Among the four studies [38,39,40,41] of acupuncture on participants with suicidal ideation, auricular acupuncture was used in three studies [39,40,41], and the NADA protocol was adopted for all the studies. In these studies, auricular acupuncture for 3–4 weeks was associated with a significant improvement in direct and indirect outcomes related to suicidal behavior, including scores of BBSI and BDI, and the salivary cortisol level. Similar results were obtained in a study [38] in which manual acupuncture was performed for 3 months, and a statistically significant reduction in BSSI and a high efficacy rate of 90.48% were reported. Among the four studies [35,36,37,43] of acupuncture on participants with depression, but suicidal behavior is not a mandatory inclusion criterion, electro-acupuncture was used in three studies [35,36,43]. Another study [37] performed auricular acupuncture, but they did not use the NADA protocol. Outcomes of suicidal behavior in the studies included item 3 of the HAMD-17 [37], NGASR [43], and CSRS [35,36]. However, the CSRS results were not reported in the literature [35,36,42], and significant improvements after 4 weeks or 1 month of acupuncture [37,43] were reported in the remaining two outcomes.

### 4.2. Clinical Implication

One notable feature of the studies on acupuncture for suicidal behavior found in this review is the frequent use of the NADA protocol (Figure 2). This protocol targets five specific auricular points, namely, the sympathetic, Shenmen, lung, kidney, and liver, and has been mainly used in the treatment of substance abuse [44,45]. Additionally, clinical evidence suggests that this treatment can be beneficial in the treatment of major depression [46] and PTSD [47]. As these mental illnesses, including major depression, substance use, and PTSD, are associated with a high risk of suicidal behavior [3,4], the NADA protocol has the potential to directly or indirectly contribute to the reduction in suicidal behavior in individuals with mental conditions. Although not included in this review as it is not an interventional study, a prospective observational study using the NADA protocol as an element of comprehensive medical and non-medical services for substance use found a statistically significant reduction in suicidal ideation (*p* = 0.005) with a 4-week comprehensive program in individuals with substance abuse [29].

The underlying therapeutic mechanisms of auricular acupuncture involving the NADA protocol include the stimulation of the vagus nerve through the auricular branches of the vagus nerve (ABVN) and anti-inflammatory and antioxidant effects [48]. In particular, the fact that auricular acupuncture stimulates the vagus nerve through ABVN means that this treatment can be regarded as a type of noninvasive vagus nerve stimulation; furthermore, it can have a beneficial effect on the stress response control [49]. Three of the included studies [39,40,41] found a significant reduction in salivary cortisol levels after auricular acupuncture in individuals with suicidal ideation, which is considered the biological basis for the improvement of the stress response. As cortisol levels are associated with suicidal behavior through hypothalamic–pituitary–adrenal axis activation [50], the findings could be regarded as a clue to explain the underlying mechanism of auricular acupuncture in suicidal behavior.

However, studies on the specific effects of auricular points and the differences in the effects of stimulation methods on suicidal behavior or related mental illness are insufficient. Among the studies included in this review, auricular acupuncture was performed using the NADA protocol [39,40,41] and auricular acupuncture using other protocols (liver, heart, spleen, kidney, Shenmen, and subcortex points) [37]; however, the effects of the two auricular acupuncture strategies could not be compared. Regarding the stimulation method, Li et al. [37] compared the effects of stimulations using magnets and vaccinia seeds on the same auricular points and found more favorable effects on the suicidal intention with the stimulation of magnets. However, since the stimulation method includes needle stimulation, electro-acupuncture, and transcutaneous electrical stimulation, the relative effect of each stimulation method on auricular acupoints needs to be further studied to investigate the optimal effects on suicidal behavior.

In addition to the physiological effects of auricular acupuncture, including the NADA protocol, there is a possibility that the treatment procedure had a positive effect on the suicidal behavior of the subjects. Specifically, the NADA protocol is typically utilized as a group therapy, where the group environment may reduce impulsiveness and enhance the enjoyment of patients [51]. Another notable auricular acupuncture procedure involves the inclusion of self-acupressure. In the study of Li et al. [37], materials for stimulation were attached to the auricular points of the participants, and the participants were instructed to stimulate themselves during the period between the treatment sessions. Self-acupressure is a type of self-care behavior that may have a positive correlation with individual self-efficacy [52,53], potentially contributing to the reduction in suicidal behavior [54]. However, the underlying mechanisms of the physiological effects of acupuncture and the potential effects of the procedure on suicidal behaviors in individuals need to be further elucidated. What is encouraging is that a 20-week clinical trial to investigate the mechanism of acupuncture in depressive patients with suicidal ideation is currently underway [55].

Although excluded because they did not meet the inclusion criteria for this review, two intervention studies of acupuncture for self-injurious behavior (SIB) [32,33] are also worth discussing. Nixon et al. [32] performed auricular acupuncture (NADA protocol) for a total of three sessions over 3 weeks for adolescent participants (n = 9) with major depressive episodes and SIB. An evaluation according to the treatment was performed at the 1-week and 4-week post-treatment follow-up. Compared to screening, acts on SIB at both 1-week and 4-week post-treatment showed a statistically significant decrease (*p* = 0.004 and *p* = 0.030, respectively). In addition, the mean internalizing anger score of the State-Trait Anger Expression Inventory significantly decreased 4 weeks post-treatment (*p* < 0.05). Participants’ satisfaction with auricular acupuncture, as reported by a brief questionnaire and clinical interviews, was high. Specifically, 86% reported that they would recommend this treatment to other individuals with SIB, and 71.4% reported that this treatment was moderate to be extremely effective. In the study of Davies et al. [33], self-administered acupuncture was recommended as an alternative coping skill for emotional distress in patients (n = 10) with emotionally unstable personality disorders with deliberate self-harm. The participants were provided with an average of two 1 h training sessions for self-administered acupuncture [33]. The participants usually inserted the needle at a nonspecific point on their arm because of the convenience of access rather than strictly stimulating the acupoint [33]. In other words, they [33] seem to regard acupuncture as an alternative to deliberate self-harming behavior. This study [33] found a dramatic decrease in weekly deliberate self-harm behavior (from 3 days per week to 0.33 days per week). As a potential alternative, self-administered acupuncture has the potential to have an increased effectiveness when included in more comprehensive self-injury prevention programs, such as the Treatment for Self-Injurious Behaviors [56].

### 4.3. Limitations

This study is the first to comprehensively investigate the effects of acupuncture on suicidal behavior. However, the following limitations of this study should be acknowledged. First, the clinical heterogeneity of the included studies prevented us from drawing firm conclusions. One reason for clinical heterogeneity is that suicidal behavior can be associated with a variety of heterogeneous conditions, including clinical and non-clinical conditions [57]. Therefore, the inclusion criteria of the population for this systematic review were not restricted; instead, studies on individuals with suicidal behavior were distinguished from studies on other individuals. If quantitative accumulation on this topic is premised in the future, it may be possible to investigate the effectiveness of acupuncture on suicidal behavior in specific clinical conditions, such as major depression. However, it should be noted that suicidal behavior still has transdiagnostic characteristics [58]. Second, the methodological quality of the included studies was suboptimal. These methodological quality deficiencies negatively affected the reliability of the findings derived from the included studies. Accordingly, the need for additional robust methodological clinical studies to confirm this preliminary finding has been emphasized. Third, the included clinical studies were insufficient to quantitatively and qualitatively understand the multifaceted effect of acupuncture on suicidal behavior. Additionally, more than half of the studies [35,36,37,38,43] included in this systematic review were conducted in China, and the others [39,40,41] were conducted in Iran. Although there are some observational studies suggesting the clinical benefits of acupuncture on suicidal behavior conducted in other countries [28,29,30,31], interventional studies targeting various ethnicities are needed to increase the possibility of a generalization. Finally, although acupuncture is considered a safe non-pharmacological treatment [59], the fact that safety data of acupuncture was reported in only three [35,36,37] of the studies included in this review is acknowledged as a methodological and ethical limitation.

### 4.4. Suggestions for Further Research

Given the findings of this review, the following future research recommendations in this field can be suggested. First, more interventional studies are needed to evaluate the therapeutic effect of acupuncture targeting individuals with suicidal behavior. In particular, studies on subjects with suicidal ideation confirmed through mental health professional clinical interviews or through some validated measures of suicidal ideation, including BSSI [21], are highly needed to evaluate the effectiveness and safety of acupuncture for suicidal behavior in the future. Second, given the importance of multidisciplinary approaches to suicide prevention, acupuncture also deserves to be investigated for its effectiveness and efficiency in other multicomponent suicide prevention programs. For example, the effectiveness of already introduced CIM services, including acupuncture, on the suicidal behavior of mentally vulnerable groups, such as veterans, could be investigated [31]. Third, some important factors, such as an individual’s experience of the acupuncture procedure, perceived effectiveness of acupuncture, and practitioner–patient communication [15], that potentially influence acupuncture procedures on suicidal behavior, could be further investigated. In addition to the importance of ethnic or cultural relevance in suicide prevention [60], it is worth investigating the acceptability, effectiveness, and experience of acupuncture procedures, a type of EATM modality, between countries familiar with and those unfamiliar with it. Finally, the safety of acupuncture for vulnerable groups, such as individuals with suicidal ideation, should be further investigated.

## 5. Conclusions

The included intervention studies found that acupuncture, including the NADA protocol, may help reduce the risk of suicidal behavior in clinical or non-clinical populations. However, owing to the low methodological quality and clinical heterogeneity of the included studies, the findings of this review cannot lead to firm conclusions. In the future, not only the effectiveness of acupuncture on the conditions but also the mediating effect related to the acupuncture procedure should be further verified.

## Figures and Tables

**Figure 1 healthcare-11-00955-f001:**
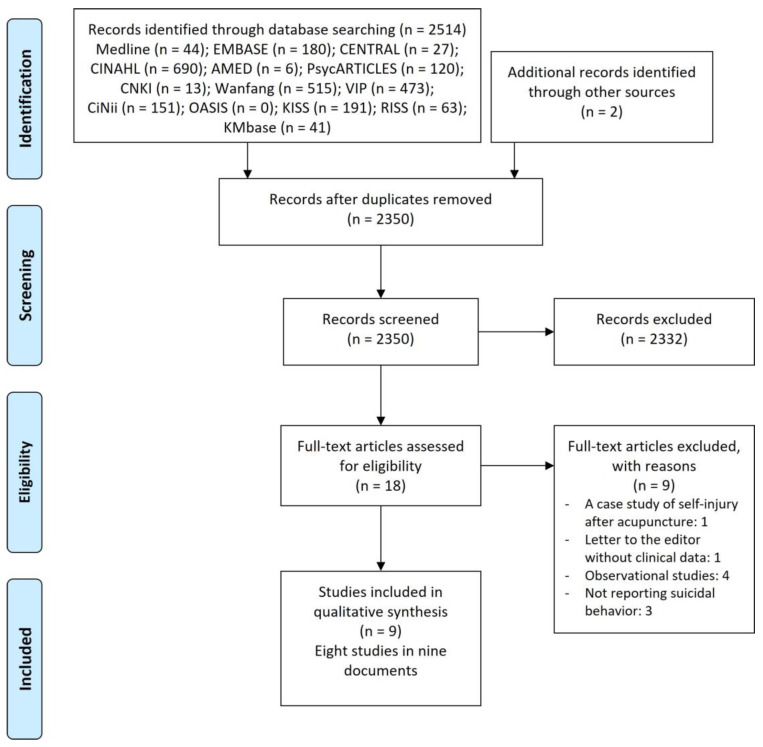
A PRISMA flow diagram of the literature screening and selection processes. Abbreviations. AMED, Allied and Complementary Medicine Database; CENTRAL, Cochrane Central Register of Controlled Trials; CINAHL, Cumulative Index to Nursing and Allied Health Literature; CNKI, China National Knowledge Infrastructure; KISS, Korean Studies Information Service System; OASIS, Oriental Medicine Advanced Searching Integrated System; RISS, Research Information Service System.

**Figure 2 healthcare-11-00955-f002:**
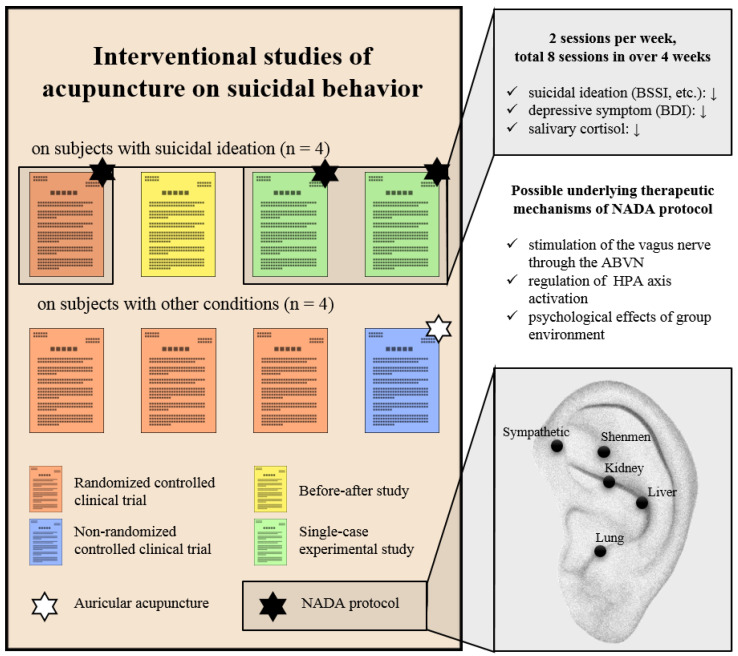
Summary of findings from this review and NADA protocol. Abbreviations. ABVN, the auricular branches of the vagus nerve; BDI, Beck Depression Inventory; BSSI, Beck Scale for Suicidal Ideation; HPA axis, hypothalamic–pituitary–adrenal axis; NADA, the National Acupuncture Detoxification Association. The down arrow sign means decrease.

**Table 1 healthcare-11-00955-t001:** Characteristics of the included studies.

Study/Country/Study Design	Sample Size (Included → Analyzed)	Clinical Condition (Diagnostic Criteria)	Intervention	Treatment Duration (f/u)	Results
Wang (2014)/China/RCT [35]	61 (20:21:20) → 55 (19:16:20)	Depressive episode (ICD), 35 ≥ HAMD-24 > 8	TG1: EA + placebo pill; TG2: EA + SSRI; CG1: sham EA + SSRI	6 weeks (4-week post-treatment)	1. CSRS
Yang (2014) and Yang et al. (2020)/China/CCT [36,42]	60 (35:25) → 55 (32:23)	Depressive episode (ICD), HAMD-24 ≥ 20	TG: EA; CG: SSRI	8 weeks (4-week post-treatment)	1. CSRS
Li et al. (2018)/China/RCT [37]	93 (31:31:31) → 83 (29:27:27)	Post-stroke depression (CCMD-3)	TG1: ROC + AA (magnetotherapy); TG2: ROC + AA (vaccaria seed); CG: ROC for stroke	4 weeks (4-week. post-treatment)	1. Item 3 of HAMD-17 (suicide)
Jin et al. (2019)/China/Before–after study [38]	21→21	Depression with suicidal ideation (CPG of depression by CMA psychiatric branch in 2015)	Acupuncture	3 months	1. CER (reduction in HAMD score); 2. Number of patients with suicidal ideation; 3. BSSI
Pirnia et al. (2019)/Iran/single-case experimental study [39]	1	Major depression with suicidal ideation, heroin abuse, and three occurrences of suicide attempts (NR)	AA + methadone	4 weeks	1. Salivary cortisol (unclear unit and the timing of sampling); 2. Suicidal ideation (unclear assessment tool)
Pirnia et al. (2019)/Iran/RCT [40]	24 (12:12) → 24 (12:12)	PDD (dysthymia) with suicidal ideation (DSM-5)	TG: AA + methadone; CG: Sham AA + methadone	4 weeks	1. Salivary cortisol sampled at random times among the three meals (unclear unit); 2. BSSI
Pirnia et al. (2020)/Iran/single-case experimental study [41]	1	Prostate carcinoma with dysthymic disorder and suicidal ideation (SCID)	AA + ROC	4 weeks	1. BDI; 2. BSSI; 3. Salivary cortisol in morning (unclear unit)
Li et al. (2021)/China/RCT [43]	90 (45:45) → 90 (45:45)	Mild and moderate depression (ICD-10), HAMD > 7	TG: EA + MT + ROC; CG: EA + ROC	1 month	1. NGASR

Abbreviations. AA, auricular acupuncture; BDI, Beck Depression Inventory; BSSI, Beck Scale for Suicidal Ideation; CCMD, Chinese Classification of Mental Disorders; CCTs, non-randomized controlled clinical trials; CER, Clinical effective rate; CG, control group; CMA, Chinese Medical Association; CPG, clinical practice guideline; CSRS, Colombia Suicide Rating Scale; DSM-IV, Diagnostic and Statistical Manual of Mental Disorders, fourth edition; EA, electro acupuncture; EUPD, emotionally unstable personality disorder; HDRS, Hamilton Depression Rating Scale; ICD, the International Classification of Disease; MT, music therapy; NGASR, Nurses Global Assessment of Suicide Risk; PDD, persistent depressive disorder; RCT, randomized controlled clinical trial; ROC, routine care; SCID, Structured Clinical Interview for DSM Disorders; SSRI, selective serotonin reuptake inhibitor; TG, treatment group.

**Table 2 healthcare-11-00955-t002:** Methodological quality of the included studies.

RCTs	Bias Arising from the Randomization Process			Bias Due to Deviations from Intended Interventions		Bias Due to Missing Outcome Data			Bias in Measurement of the Outcome			Bias in Selection of the Reported Result		Overall Bias		
Wang (2014) [35]	Low risk			Some concerns		Low risk			Low risk			Some concerns		Some concerns		
Li et al. (2018) [37]	Low risk			Some concerns		Some concerns			High risk			Some concerns		High risk		
Pirnia et al. (2019) [40]	Some concerns			Some concerns		Low risk			Low risk			Some concerns		Some concerns		
Li et al. (2021) [43]	Some concerns			Some concerns		Low risk			High risk			Some concerns		High risk		
CCTs	Selection bias due to the selection of inappropriate comparison target group		Selection bias due to inappropriate intervention or inappropriate selection of exposure group or patient group		Selection bias due to inappropriate confounder confirmation and consideration		Performance bias due to inappropriate intervention or inappropriate exposure measurement		Confirmation bias due to inappropriate blinding of assessors		Confirmation bias due to inappropriate outcome assessment methods		Attrition bias due to inappropriate handling of incomplete data		Reporting bias due to selective outcome reporting	
Yang (2014) and Yang et al. (2020) [36,42]	Low risk		High risk		High risk		Low risk		Low risk		Low risk		Unclear		Low risk	
Before–after studies	Q1	Q2		Q3	Q4	Q5		Q6	Q7	Q8		Q9	Q10	Q11		Q12
Jin et al. (2019) [38]	Yes	Yes		Yes	CD	No		Yes	Yes	No		Yes	Yes	Yes		NA

Abbreviations. CCTs, non-randomized controlled clinical trials; CD, cannot determine; NA, not applicable; RCT, randomized controlled clinical trial. Note. Q1. Was the study question or objective clearly stated? Q2. Were eligibility/selection criteria for the study population prespecified and clearly described? Q3. Were the participants in the study representative of those who would be eligible for the test/service/intervention in the general or clinical population of interest? Q4. Were all eligible participants that met the prespecified entry criteria enrolled? Q5. Was the sample size sufficiently large to provide confidence in the findings? Q6. Was the test/service/intervention clearly described and delivered consistently across the study population? Q7. Were the outcome measures prespecified, clearly defined, valid, reliable, and assessed consistently across all study participants? Q8. Were the people assessing the outcomes blinded to the participants’ exposures/interventions? Q9. Was the loss to follow-up after baseline 20% or less? Were those lost to follow-up accounted for in the analysis? Q10. Did the statistical methods examine changes in outcome measures from before to after the intervention? Were statistical tests done that provided *p* values for the pre-to-post changes? Q11. Were outcome measures of interest taken multiple times before the intervention and multiple times after the intervention (i.e., did they use an interrupted time-series design)? Q12. If the intervention was conducted at a group level (e.g., a whole hospital, a community, etc.) did the statistical analysis take into account the use of individual-level data to determine effects at the group level?

**Table 3 healthcare-11-00955-t003:** Details of acupuncture procedures among included studies.

Study	Intervention	Acupoint	Depth of Insertion	Response Sought	Needle Stimulation	Needle Retention Time	Needle Type	Practitioner Background	Frequency (Treatment Duration)/Number of Sessions	Concurrent Treatment
Wang (2014) [35]	TG1: EA + placebo pill; TG2: EA + SSRI; CG1: placebo EA + SSRI	ST36, SP6, PC6, LR3, HT7 (additional 2–3 additional points based on clinical symptoms)	0.5–0.8 cun	De qi	1–50 Hz, disperse-dense waves, −10 mA depending on the participant’s comfort level	30 min	0.30 × 25 mm and 0.30 × 40 mm	NR	3 sessions per week (6 weeks)/18	SSRI (escitalopram)
Yang (2014) and Yang et al. (2020) [36,42]	TG: EA; CG: SSRI	GV20, EX-HN3 (additional 2–5 additional points based on TCM syndrome differentiation)	0.5–0.8 cun	NR	1–50 Hz, disperse-dense waves, 0.1–5 mA depending on the participant’s comfort level	30 min	0.32 mm	NR	3 sessions per week (8 weeks)/24	None
Li et al. (2018) [37]	TG1 and 2: ROC + AA; CG: ROC for stroke	(Unilateral) liver, heart, spleen, kidney, Shenmen, subcortex	NA	NA	Self-acupressure	3 days	TG1: magnetotherapy TG2: vaccaria seed	NR	3 sessions per week (4 weeks)/12	ROC for stroke (e.g., blood pressure control, cerebral edema management, and rehabilitation training)
Jin et al. (2019) [38]	Acupuncture	(bilateral) SP4, PC6, BL62, SI3, GB41, TE5, LU7, KI6, BL15, BL18, BL23, CV12, ST25, CV6, CV4, KI3, HT7, LR3	Back-su point: 1–1.2 cun; acupoints of the arms and legs: 0.5–1 cun (except for HT7: 0.3–0.5 cun); Abdominal acupoints: 1–1.2 cun	No De-qi	NR	40 min	0.25 × 40 mm	NR	1st month: 5 sessions per week; 2nd and 3rd months: 3 sessions per week/44	Infrared irradiation
Pirnia et al. (2019) [39]	AA + methadone	(Bilateral) sympathetic, Shenmen, kidney, liver, lung (NADA protocol)	2–3 mm	NR	NR	NR	0.25 × 13 mm	NR	2 sessions per week (4 weeks)/8	Methadone syrup 8 mg/d
Pirnia et al. (2019) [40]	TG: AA + methadone; CG: Sham AA + methadone	(Bilateral) sympathetic, Shenmen, kidney, liver, lung (NADA protocol)	2–3 mm	NR	NR	NR	0.25 × 13 mm	NR	2 sessions per week (4 weeks)/8	Methadone syrup 35 ± 0.3 cc/d
Pirnia et al. (2020) [41]	AA + ROC	(Bilateral) sympathetic, Shenmen, kidney, liver, lung (NADA protocol)	2–3 mm	NR	NR	30–45 min	0.25 × 13 mm	a trained physician and an acupuncturist with a certification and five-year history of the treatment	2 sessions per week (4 weeks)/8	ROC (degarelix 40 mg/d, venlafaxine 225 mg/d, and methadone syrup 15 mg/d)
Li et al. (2021) [43]	TG: EA + MT + ROC; CG: EA + ROC	GV20, PC6, HT7, LI4, CV17, ST36, SP6 (optional: CV12, LR3, LR2, GB40)	NR	NR	1.5 Hz, gradually increase the voltage until the skin next to the acupoints of the patient twitches slightly	30 min	NR	NR	1 session per day (1 month)/30	MT + ROC (psychological support, health education, etc.)

Abbreviations. AA, auricular acupuncture; CG, control group; EA, electro acupuncture; MT, music therapy; NA, not applicable; NADA, the National Acupuncture Detoxification Association; NR, not reported; ROC, routine care; SSRI, selective serotonin reuptake inhibitor; TG, treatment group.

## Data Availability

This data used to support the findings of this study are included within the article.

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
