# Peer review of "The Effectiveness and Safety of Acupuncture on Suicidal Behavior: A Systematic Review"

_healthcare, 2023, doi:10.3390/healthcare11070955_

Round 1

Reviewer 1 Report

The manuscript deals with a systematic review of the effect of acupuncture on suicidal behavior.

The most significant limitation of the work is the inclusion of self-injurious behaviors as part of suicidal behavior, which is not correct. It is necessary to give a precise definition of suicidal behavior. Although self-injurious behaviors are a risk factor for suicidal behavior, it is not the only risk factor. Moreover, studies that assess self-injurious behaviors do not assess suicidal behaviors as outcomes in their methodological designs.

Therefore, to comply with what is declared in the title and objective of the manuscript, it should be limited only to those studies that evaluate suicidal ideation, suicide attempt, or suicide.

In terms of the safety of acupuncture for treating suicidal behaviors, the failure to report data on the safety of the method is a methodological and ethical limitation in these studies.

In my opinion, it is necessary to limit the review only to suicidal ideation, assessed by valid instruments or through a clinical interview with a mental health professional.

Author Response

  • Response to Comments from Reviewer 1

Comment 1:

The manuscript deals with a systematic review of the effect of acupuncture on suicidal behavior.

The most significant limitation of the work is the inclusion of self-injurious behaviors as part of suicidal behavior, which is not correct. It is necessary to give a precise definition of suicidal behavior. Although self-injurious behaviors are a risk factor for suicidal behavior, it is not the only risk factor. Moreover, studies that assess self-injurious behaviors do not assess suicidal behaviors as outcomes in their methodological designs.

Therefore, to comply with what is declared in the title and objective of the manuscript, it should be limited only to those studies that evaluate suicidal ideation, suicide attempt, or suicide.

Response:

Thank you for your review. As the reviewer comments, self-harm can be considered an important risk factor for suicidal behavior, but it is difficult to be included in suicidal behaviors. Accordingly, we excluded the two studies of self-harm from this systematic review. However, because self-harm is an important risk factor for suicidal behavior, the two excluded studies are mentioned in the Discussion section.

- Nixon, M.K.; Cheng, M.; Cloutier, P. An open trial of auricular acupuncture for the treatment of repetitive self-injury in depressed adolescents. The Canadian child and adolescent psychiatry review 2003, 12, 10-12.

- Davies, S.; Bell, D.; Irvine, F.; Tranter, R. Self-administered acupuncture as an alternative to deliberate self-harm: a feasibility study. J Pers Disord 2011, 25, 741-754, doi:10.1521/pedi.2011.25.6.741.

“Although excluded because they did not meet the inclusion criteria for this review, two intervention studies of acupuncture for self-injurious behavior (SIB) [32,33] are also worth discussing. Nixon et al. [32] performed auricular acupuncture (NADA protocol) for a total of three sessions over 3 weeks for adolescent participants (n = 9) with major depressive episodes and SIB. Evaluation according to treatment was performed at 1-week and 4-week post-treatment follow-up. Compared to screening, acts on SIB at both 1-week and 4-week post-treatment showed a statistically significant decrease (p = 0.004 and p = 0.030, respectively). In addition, the mean internalizing anger score of the State-Trait Anger Expression Inventory significantly decreased at 4-week post-treatment (p < 0.05). Participants' satisfaction with auricular acupuncture, as reported by a brief questionnaire and clinical interviews, was high. Specifically, 86% reported that they would recommend this treatment to other individuals with SIB, and 71.4% reported that this treatment was moderate to extremely effective. In the study of Davies et al. [33], self-administered acupuncture was recommended as an alternative coping skill for emotional distress in patients (n = 10) with emotionally unstable personality disorders with deliberate self-harm. The participants were given an average of two 1-h training sessions for self-administered acupuncture [33]. The participants usually inserted the needle at a nonspecific point on their arm because of the convenience of access rather than strictly stimulating the acupoint [33]. In other words, they [33] seem to regard acupuncture as an alternative to deliberate self-harming behavior. This study [33] found a dramatic decrease in weekly deliberate self-harm behavior (from 3 days per week to 0.33 days per week). As a potential alternative, self-administered acupuncture has the potential for increased effectiveness when included in more comprehensive self-injury prevention programs, such as the Treatment for Self-Injurious Behaviors [56].”

(Please refer to page 14, red words)

Comment 2:

In terms of the safety of acupuncture for treating suicidal behaviors, the failure to report data on the safety of the method is a methodological and ethical limitation in these studies.

Response:

Thank you for your review. We agree with the reviewer's comments, and have added methodological and ethical limitations related to the poorly reported safety profile of acupuncture to the Discussion section.

Finally, although acupuncture is considered a safe non-pharmacological treatment [59], the fact that safety data of acupuncture was reported in only three [35-37] of the studies included in this review is acknowledged as a methodological and ethical limitation.”

(Please refer to page 15, red words)

Comment 3:

In my opinion, it is necessary to limit the review only to suicidal ideation, assessed by valid instruments or through a clinical interview with a mental health professional.

Response:

Thank you for your review. The reviewer's comments are considered highly clinically relevant. However, given the lack of research in this topic, it seems that the valuable comments can be considered as suggestions for further research in this area. We added the following considerations.

“Given the findings of this review, the following future researches in this field can be suggested. First, more interventional studies are needed to evaluate the therapeutic effect of acupuncture targeting individuals with suicidal behavior. In particular, studies on subjects with suicidal ideation confirmed through mental health professional clinical interviews or through some validated measures of suicidal ideation including BSSI [21] are highly needed to evaluate the effectiveness and safety of acupuncture for suicidal behavior in the future.”

(Please refer to page 15, red words)

Reviewer 2 Report

Please read the attachment. Thank you.

Author Response

  • Response to Comments from Reviewer 2

Comment 1:

Please read the attachment. Thank you.

- Explain how to select a survey sample. The authors identified 2514 articles for survey and meta-analysis based on the WHAT formula.

Response:

Thank you for your review. We additionally described the process from the 2514 documents initially retrieved to the final selection of documents to be included in this review in more detail (i.e., the two-step selection process).

“In the initial search, 2516 articles were retrieved, including additional two records identified through other sources. In the first step, the titles and abstracts of 2350 articles were reviewed, excluding duplicate articles. During the initial screening process, 2332 articles evaluated as not relevant to this systematic review were excluded. In the second step, through a full-text review, nine articles that do not meet the inclusion criteria were excluded. The reasons for the papers excluded in the second step are as follows: a case study of self-injury after acupuncture [26], a letter to the editor without clinical data [27], four observational studies [28-31], and three not reporting suicidal behaviors as their outcomes [32-34]. Finally, eight studies in nine articles [35-43] were included in this review by the two-step selection process. As Yang [36] (dissertation) and Yang et al. [42] (journal article) reported the same study, only the former [36] will be cited; however, some of its data comes from the latter [42]. The study selection process is illustrated in Figure 1.”

(Please refer to page 4, red words)

Comment 2:

- It is necessary to specify some examples of some common reasons leading to suicide.

Response:

Thank you for your review. We have added examples of common causes of suicide.

“Worldwide, suicide accounts for 1.4% of all premature deaths, the majority of which are related to mental illnesses, particularly depressive disorders, substance use, psychosis, post-traumatic stress disorder (PTSD), and impulse control [3,4]. However, risk factors for suicide are not limited to mental illness. For example, some physical health conditions such as human immunodeficiency virus infection and acquired immunodeficiency syndrome, sleep disorders, and traumatic brain injury are also associated with a high risk of suicide [5]. Additionally, social context, including unjust and discriminatory social conditions and attitudes, may contribute to an individual's suicide [6]. Common methods of suicide attempt include hanging, self-poisoning with pesticides, and the use of firearms [3]. Therefore, suicide prevention should consider complex factors at both individual and social levels.”

(Please refer to pages 1-2, red words)

Comment 3:

Are the statistics of the death rate by country required to warn and influence the government of that country to take timely intervention measures and policies? If necessary, should we add it to positively impact the government's policy-making, the world health organization, the ministry of education and training, and related ministries with programs appropriate to national policy?

Response:

Thank you for your review. In suicide prevention, as the reviewer commented, a societal and national approach, policymaking, is important. We added the following comments.

“Also, given the differences in mortality statistics across countries and the importance of developing programs suited to each country's policy [3], it may be more relevant in countries where the use of either the EATM or CIM modality is common.”

(Please refer to page 13, Figure 2)

Comment 4:

- There are explanations and discussions in words. Numbers and statistics should be shown in figures and tables that will better understand and attract many future readers.

Response:

Thank you for your review. We have added Figure 2 to aid future readers' understanding.

(Please refer to page 2, red words)

Comment 5:

- Please suggest research trends of the fields and aspects that need to be studied more.

Response:

Thank you for your review. Based on the reviewer's comment, ‘4.4. Suggestions for further research’ was added in the Discussion section.

4.4. Suggestions for further research

Given the findings of this review, the following future researches in this field can be suggested. First, more interventional studies are needed to evaluate the therapeutic effect of acupuncture targeting individuals with suicidal behavior. In particular, studies on subjects with suicidal ideation confirmed through mental health professional clinical interviews or through some validated measures of suicidal ideation including BSSI [21] are highly needed to evaluate the effectiveness and safety of acupuncture for suicidal behavior in the future. Second, given the importance of multidisciplinary approaches to suicide prevention, acupuncture also deserves to be investigated for its effectiveness and efficiency in other multicomponent suicide prevention programs. For example, the effectiveness of already introduced CIM services including acupuncture on suicidal behavior of mentally vulnerable groups such as veterans could be investigated [31]. Third, some important factors such as individual’s experience of the acupuncture procedure, perceived effectiveness of acupuncture, and practitioner-patient communication [15] that potentially influence acupuncture procedures on suicidal behavior could be further investigated. In addition to the importance of ethnic or cultural relevance in suicide prevention [60], it is worth investigating the acceptability, effectiveness, and experience of acupuncture procedures, a type of EATM modality, between countries familiar with and those unfamiliar with it. Finally, the safety of acupuncture for vulnerable groups such as individuals with suicidal ideation should be further investigated.”

(Please refer to page 15, red words)

Comment 6:

- Propose measures to support directly from the psychophysiological treatment program from a medical perspective if possible.

Response:

Thank you for your review. In the ‘4.4. Suggestions for further research’, we added the following sentences.

Second, given the importance of multidisciplinary approaches to suicide prevention, acupuncture also deserves to be investigated for its effectiveness and efficiency in other multicomponent suicide prevention programs. For example, the effectiveness of already introduced CIM services including acupuncture on suicidal behavior of mentally vulnerable groups such as veterans could be investigated [31].”

(Please refer to page 15, red words)

Comment 7:

- References best used within five years will reflect the current state of the research problem involved. Older documents may not be a reliable reference source or inappropriate, and the authors should note this point and update their references if possible.

Response:

Thank you for your review. The following references have been updated.

Woodward, A.T.; Bullard, K.M.; Taylor, R.J.; Chatters, L.M.; Baser, R.E.; Perron, B.E.; Jackson, J.S. Complementary and alternative medicine for mental disorders among African Americans, black Caribbeans, and whites. Psychiatr Serv 2009, 60, 1342-1349

→ Ashraf, H.; Salehi, A.; Sousani, M.; Sharifi, M.H. Use of Complementary Alternative Medicine and the Associated Factors among Patients with Depression. Evid Based Complement Alternat Med 2021, 2021, 6626394

Miyazaki, S.; Hagihara, A.; Mukaino, Y. Acupuncture practitioner-patient communication in Japan. International journal of general medicine 2008, 1, 83-90

→ Prady, S.L.; Burch, J.; Crouch, S.; MacPherson, H. Controlling practitioner-patient relationships in acupuncture trials: a systematic review and meta-regression. Acupunct Med 2013, 31, 162-171

Comment 8:

The reviewer hopes that his point of view could help the authors improve their work. I could not find any significant changes to this manuscript, and I appreciate your work.

Sincerely yours,

Response:

Thank you for your review.

Reviewer 3 Report

This is innovative and useful. I understand why the authors had to deviate from the published protocol; was a revised protocol published?

lines 212-214 - might the CARE guidelines have been used here?

Could themed subsectioning be considered in the discussion/conclusion?

Minor corrections to English language suggested:

line 12 - 'used for long' should read 'used for a long time'

line 83 - 'electronic databases for which' should read 'electronic databases in which'

line 109 - 'or on the stimulation method' - clearer just to say 'penetrated the skin or not'

line 138 - 'the results' should just read 'results'

line 142 - just 'resolved through discussion' (as at line 131)

line 281 - perhaps rephrase 'A significant main effect of time'

lines 282, 284 and 355 - 'acts on SIB' should perhaps read 'acts of SIB' or 'acting on SIB'

lines 304-5 - 'Eight sessions of auricular acupuncture (NADA protocol) were performed for 4 weeks on the patient' - can you make this clearer, ie; 2 txs x week?

lines 363-4 - unclear - can this be rephrased? - 'not a mandatory inclusion criterion, electroacupuncture was used in two studies'

lines 380-382 - 'vagus nerve' and 'vagal nerve' - for consistency, use the same term every time here

line 424 - an extra [31] mid-line could be removed

line 428 - 'to increase its effectiveness by being included' could be rephrased as 'for increased effectiveness when included

lines 433-4 - 'can be referred [29]. This study found' - could just say 'found' here

line 447 'on the future' should read 'in the future'

Could Supplementary Material links be titled 'Supplementary Material' rather than just 'Supplementary'?

Author Response

  • Response to Comments from Reviewer 3

Comment 1:

This is innovative and useful. I understand why the authors had to deviate from the published protocol; was a revised protocol published?

Response:

Thank you for your careful review. We excluded two studies that differed from the published protocol. That is, the following two studies that reported the effect of acupuncture on self-injury were excluded. This is because, self-harm is a risk factor for suicidal behavior, but it is not included in suicidal behavior.

- Nixon, M.K.; Cheng, M.; Cloutier, P. An open trial of auricular acupuncture for the treatment of repetitive self-injury in depressed adolescents. The Canadian child and adolescent psychiatry review 2003, 12, 10-12.

- Davies, S.; Bell, D.; Irvine, F.; Tranter, R. Self-administered acupuncture as an alternative to deliberate self-harm: a feasibility study. J Pers Disord 2011, 25, 741-754, doi:10.1521/pedi.2011.25.6.741.

However, because self-harm is an important risk factor for suicidal behavior, the two excluded studies are mentioned in the Discussion section.

“Although excluded because they did not meet the inclusion criteria for this review, two intervention studies of acupuncture for self-injurious behavior (SIB) [32,33] are also worth discussing. Nixon et al. [32] performed auricular acupuncture (NADA protocol) for a total of three sessions over 3 weeks for adolescent participants (n = 9) with major depressive episodes and SIB. Evaluation according to treatment was performed at 1-week and 4-week post-treatment follow-up. Compared to screening, acts on SIB at both 1-week and 4-week post-treatment showed a statistically significant decrease (p = 0.004 and p = 0.030, respectively). In addition, the mean internalizing anger score of the State-Trait Anger Expression Inventory significantly decreased at 4-week post-treatment (p < 0.05). Participants' satisfaction with auricular acupuncture, as reported by a brief questionnaire and clinical interviews, was high. Specifically, 86% reported that they would recommend this treatment to other individuals with SIB, and 71.4% reported that this treatment was moderate to extremely effective. In the study of Davies et al. [33], self-administered acupuncture was recommended as an alternative coping skill for emotional distress in patients (n = 10) with emotionally unstable personality disorders with deliberate self-harm. The participants were given an average of two 1-h training sessions for self-administered acupuncture [33]. The participants usually inserted the needle at a nonspecific point on their arm because of the convenience of access rather than strictly stimulating the acupoint [33]. In other words, they [33] seem to regard acupuncture as an alternative to deliberate self-harming behavior. This study [33] found a dramatic decrease in weekly deliberate self-harm behavior (from 3 days per week to 0.33 days per week). As a potential alternative, self-administered acupuncture has the potential for increased effectiveness when included in more comprehensive self-injury prevention programs, such as the Treatment for Self-Injurious Behaviors [56].”

(Please refer to page 14, red words)

Comment 2:

lines 212-214 - might the CARE guidelines have been used here?

Response:

Thank you for the comment. Because this systematic review did not include case reports, the CARE guideline was not used.

Comment 3:

Could themed subsectioning be considered in the discussion/conclusion?

Response:

Thank you for the comment. Subsections of the Discussion section have been added as follow.

“4.1. Summary of findings

4.2. Clinical implication

4.3. Limitations

4.4. Suggestions for further research”

(Please refer to pages 12-15, red words)

Comment 4:

Minor corrections to English language suggested:

line 12 - 'used for long' should read 'used for a long time'

line 83 - 'electronic databases for which' should read 'electronic databases in which'

line 109 - 'or on the stimulation method' - clearer just to say 'penetrated the skin or not'

line 138 - 'the results' should just read 'results'

line 142 - just 'resolved through discussion' (as at line 131)

line 281 - perhaps rephrase 'A significant main effect of time'

lines 282, 284 and 355 - 'acts on SIB' should perhaps read 'acts of SIB' or 'acting on SIB'

lines 304-5 - 'Eight sessions of auricular acupuncture (NADA protocol) were performed for 4 weeks on the patient' - can you make this clearer, ie; 2 txs x week?

lines 363-4 - unclear - can this be rephrased? - 'not a mandatory inclusion criterion, electroacupuncture was used in two studies'

lines 380-382 - 'vagus nerve' and 'vagal nerve' - for consistency, use the same term every time here

line 424 - an extra [31] mid-line could be removed

line 428 - 'to increase its effectiveness by being included' could be rephrased as 'for increased effectiveness when included

lines 433-4 - 'can be referred [29]. This study found' - could just say 'found' here

line 447 'on the future' should read 'in the future'

Response:

Thank you for your kind comments. The manuscript has been revised to reflect all of the reviewer's comments and has been marked in red.

Comment 5:

Could Supplementary Material links be titled 'Supplementary Material' rather than just 'Supplementary'?

Response:

Thank you for the comment. ‘Supplementary’ was modified as ‘Supplementary Material’.

Round 2

Reviewer 1 Report

The authors followed the suggestions made in the first review.

 The manuscript has the necessary elements for its publication in its current condition.